# Inhibition of *Asaia* in Adult Mosquitoes Causes Male-Specific Mortality and Diverse Transcriptome Changes

**DOI:** 10.3390/pathogens9050380

**Published:** 2020-05-15

**Authors:** Maria Vittoria Mancini, Claudia Damiani, Sarah M. Short, Alessia Cappelli, Ulisse Ulissi, Aida Capone, Aurelio Serrao, Paolo Rossi, Augusto Amici, Cristina Kalogris, George Dimopoulos, Irene Ricci, Guido Favia

**Affiliations:** 1School of Biosciences and Veterinary Medicine, University of Camerino, 62032 Camerino, Italy; MariaVittoria.Mancini@glasgow.ac.uk (M.V.M.); claudia.damiani@unicam.it (C.D.); alessia.cappelli@unicam.it (A.C.); ulisse.ulissi@unicam.it (U.U.); aida.capone@unicam.it (A.C.); aurelio.serrao@unicam.it (A.S.); paolo.rossi@unicam.it (P.R.); augusto.amici@unicam.it (A.A.); cristina.kalogris@unicam.it (C.K.); irene.ricci@unicam.it (I.R.); 2MRC-University of Glasgow-Centre for Virus Research, Glasgow G61 1QH, UK; 3Centro Interuniversitario di Ricerca sulla Malaria (Italian Malaria Network), Italy; 4Department of Entomology, The Ohio State University, Columbus, OH 43210, USA; short343@osu.edu; 5W. Harry Feinstone Department of Molecular Microbiology and Immunology, Bloomberg School of Public Health, Johns Hopkins University, Baltimore, MD 21205, USA; gdimopo1@jhu.edu

**Keywords:** *Asaia*, *Anopheles*, symbiont

## Abstract

Mosquitoes can transmit many infectious diseases, such as malaria, dengue, Zika, yellow fever, and lymphatic filariasis. Current mosquito control strategies are failing to reduce the severity of outbreaks that still cause high human morbidity and mortality worldwide. Great expectations have been placed on genetic control methods. Among other methods, genetic modification of the bacteria colonizing different mosquito species and expressing anti-pathogen molecules may represent an innovative tool to combat mosquito-borne diseases. Nevertheless, this emerging approach, known as paratransgenesis, requires a detailed understanding of the mosquito microbiota and an accurate characterization of selected bacteria candidates. The acetic acid bacteria *Asaia* is a promising candidate for paratransgenic approaches. We have previously reported that *Asaia* symbionts play a beneficial role in the normal development of *Anopheles* mosquito larvae, but no study has yet investigated the role(s) of *Asaia* in adult mosquito biology. Here we report evidence on how treatment with a highly specific anti-*Asaia* monoclonal antibody impacts the survival and physiology of adult *Anopheles stephensi* mosquitoes. Our findings offer useful insight on the role of *Asaia* in several physiological systems of adult mosquitoes, where the influence differs between males and females.

## 1. Introduction

Mosquito-borne diseases, and more generally vector-borne diseases, pose serious threats to public health; about 700,000 people die each year due to direct or indirect consequences of them [1]. There are no effective vaccines for many of these diseases (e.g., malaria or viral fevers) and in some cases, drugs are losing efficacy due to pathogen and vector resistance. Consequently, innovative control methods targeting the mosquito vector are urgently required. 

Paratransgenesis is an increasingly intriguing approach in which bacteria are genetically modified to secrete anti-pathogen molecules and introduced into mosquito vectors. These bacteria then interfere with vector competence or with important host physiological traits (e.g., reproduction, development).

Compared to other transmission blocking-systems (e.g., genetically modified mosquitoes), paratransgenesis is considered potentially more tractable. This is in part because effector molecule(s) expressed by symbiotic bacteria could be easily replaced by more novel and effective ones when efficacy is reduced, since the logistics of producing bacteria on a large scale is much simpler than the mass-rearing of genetically modified mosquitoes for population replacement [2]. Furthermore, in many endemic areas, vector competence for the same pathogen is shared among various mosquito species, whose modification would be extremely challenging. This impairment would be overcome by using a unique modified bacterium able to stably invade several vectors. Paratransgenesis may therefore become an important tool for the control of mosquito-borne diseases and, more generally, vector control [3].

To successfully implement this approach, a detailed, system-based understanding of the microbiota and its interactions with the host is critical. Additionally, bacterial features allowing mutualism and persistence within the host ecosystem and the mechanisms regulating and balancing such associations need to be elucidated in order to provide essential information for an effective application. Among other members of the microbial communities naturally inhabiting mosquito tissues, *Asaia* spp. have shown potentially exploitable characteristics. This acetic acid bacterium is one of the most promising mosquito symbionts for the paratransgenic approach [4]. Large-scale studies on the ability of *Asaia* to invade populations, combined with the recent construction of a recombinant *Asaia* strain expressing anti-*Plasmodium* effector molecules, strongly support the use of paratransgenic *Asaia* strains in the field [5,6]. We have also shown that *Asaia* may activate the basal level of mosquito immunity, thus interfering with development of the malaria parasite in *Anopheles stephensi* [7]. Immune gene stimulation by *Asaia* is not limited to mosquitoes, as a similar phenomenon was also demonstrated in leafhoppers [8].

We have also shown that *Asaia* is able to accelerate the developmental transition from larvae to adults in *An. stephensi* [9,10]. Nevertheless, despite this very promising progress, not much is known about the physiological role(s) exerted by this bacterium on the mosquito host. 

Effort toward the field application of *Asaia* must concurrently be focused on a comprehensive understanding of the function of this symbiont within its mosquito hosts. 

We hypothesized that *Asaia* provides a fitness benefit during the adult stage. To address this, we performed a loss-of-function analysis using a highly specific anti-*Asaia* monoclonal antibody (mAb), intra-thoracically injected into *An. stephensi* mosquitoes, to identify possible phenotypic alterations and thereby investigate the effect(s) that *Asaia* may exert on adult mosquitoes.

## 2. Results and Discussion

The specificity of the anti-*Asaia* mAb was assessed through a double-step approach. First, we performed an immunofluorescence assay (IFA) on *An. stephensi* mosquitoes injected with the anti-*Asaia* mAb, showing that the mAb recognized only rod-like bacteria (Figure 1). No unspecific hybridization was observed on tissues. In particular, specific fluorescent signals in male guts were observed (Figure 1B,D,E). Similar specificity was detected in the crop and ovaries of females (Appendix A).

Moreover, to further prove that the mAb specifically recognizes *Asaia* and no other components of the mosquito microbiota, an additional IFA was performed on *An. stephensi* mosquitoes orally colonized with an *Asaia* strain expressing the green fluorescent protein (named *Asaia*-GFP) [5], and then intra-thoracically injected with anti-*Asaia* antibody. The red signal from the anti-*Asaia* mAb co-localized with *Asaia*-GFP on midguts, suggesting a specific binding with *Asaia* and ruling-out a possible cross reaction with other bacteria species or host antigens (Figure 2).

Once the specificity of the mAb was confirmed, it was administered to different cohorts of *An. stephensi*. Three treatments were tested on adult *An. stephensi* males and females: (1) injection of anti-*Asaia* mAb, (2) injection of anti-Herceptin mAb (Herc is a humanized monoclonal antibody used against advanced breast cancer that has relapsed or spread to other organs, here used as an unspecific control), or (3) mock injection with PBS, and their survival after injection was monitored every day. 

We observed a significant sex-biased life-shortening effect (Figure 3A–C), where males appeared to be particularly affected: within three days from the anti-*Asaia* mAb administration all male mosquitoes had died (Figure 3A), while control-injected mosquitoes treated with Herc (Figure 3D) or PBS (Figure 3E) showed a significantly longer survival (*p* < 0.0001). In fact, the mortality rates in control mosquitoes (PBS and Herc) were 50% at day 16 and 100% at day 35.

In contrast, the effect of anti-*Asaia* mAb on adult females was moderate. Females injected with anti-*Asaia* mAb showed a mortality rate of 50% at day 23 and 99% at day 30, consistent with control rates; those treated with PBS had a mortality rate of 50% at day 26 and 88% at day 30, while those injected with the Herc mAb had a mortality rate of 50% at day 26 and 99% at day 30 (Figure 3B).

The remarkable observed sex-specificity of anti-*Asaia* mAb treatment suggests that *Asaia* could have a substantial influence on the longevity of *An. stephensi* mosquitoes, particularly in males. Although the mechanisms explaining how the anti-*Asaia* mAb affects *Asaia* cells or the bacterial population remain unclear, a potential interference with the basic physiological functions of *Asaia* (e.g., binding, aggregation) in the digestive tract or a more direct effect causing bacterial cell death could be hypothesized [11]. Additionally, we cannot rule out the possibility of off-target effects on microbes outside the digestive tract and reproductive tract, though we note that previous work has shown that *Asaia* is rarely found outside these tissues in *An. stephensi*. We also cannot eliminate the possibility that the anti-*Asaia* mAb could be affecting male tissues in a way that we were unable to measure here. However, we note that treatment with a control antibody (Herc) had no effect on male longevity, suggesting that antibody treatment per se is not detrimental to male survival. There are many potential mechanisms by which perturbation of the *Asaia* community may be exerting an influence on host longevity. *Asaia* could provide critical nutrient(s) to the mosquitoes or be involved in nutrient uptake or detoxification, although the discovery of *An. stephensi* populations lacking *Asaia* in their natural microbiota [12] suggests that the *Asaia*–*An. stephensi* symbiosis could have different traits depending on bacterial variants and host backgrounds. Alternatively, *Asaia* could provide beneficial effects, possibly protecting against pathogenic microbes, interfering with the colonization of other bacteria, or inducing immunological or barrier defenses in the mosquito gut [7,8].

The observation of a sex-biased life-shortening led us to further investigate the molecular basis of this phenotype with a transcriptomic analysis. A genome-wide microarray-based transcriptome comparison between male and female mosquitoes either injected with anti-*Asaia* mAb or with Herc antibody was performed.

As expected, transcriptomic findings suggested sex-differentiated regulation of several genes (Figure 4). 

Compared to controls, in anti-*Asaia* mAb treated males we observed a profound alteration in gene expression: 79 genes were differentially regulated, among which only 9 are shared with females injected with anti-*Asaia* mAb (Figure 4A). Similarly, the overall profile of females injected with anti-*Asaia* mAb showed changes in the transcript abundance of 27 genes, 18 of which are female-specific (Figure 4A). Sixteen female-specific genes (88.9%) were up-regulated in response to treatment, while only two (11.1%) were down-regulated. Forty of the male-specific genes (57.1%) were up-regulated in response to treatment while 30 (42.9%) were down-regulated. We also identified nine up-regulated common genes between males and females treated with anti-*Asaia* mAb. Two genes (ASTE008182 and ASTE009034) were sex specific at the probe level but shared at the gene level, and were therefore excluded from analysis. 

Among the anti-*Asaia* mAb treated male-specific transcripts, 10 corresponded to genes predicted to have serine-type endopeptidase activity. Molecular function Gene Ontology (GO) analysis of the male-specific gene list revealed a significant over-representation of genes belonging to the GO term GO:0004252, “Serine Type Endopeptidase Activity” (p = 8.19 × 10^−5^), as well as the parent terms “Peptidase Activity” (GO:0008233, p = 0.0045) and “Catalytic Activity” (GO:0003824, p = 0.0066) (Figure 4B, Appendix A). Genes in these categories included multiple CLIP domain serine proteases, cytochrome P450s, and peptidoglycan recognition proteins. CLIP domain serine proteases are known to be involved in the insect immune response, regulating immune system signaling and melanization [13]. In mosquitoes, CLIP proteases have been shown to play a role in the immune response, and especially in the anti-*Plasmodium* complement-like pathway and melanization [14,15]. Peptidoglycan recognition proteins (PGRPs) are critical for the detection of pathogen-associated molecular patterns, leading to the activation of anti-bacterial responses in insects, and in mosquitoes, PGRPs play a key role in defense against *Plasmodium* parasites [16]. These results suggest that the inhibition of *Asaia* in males results in altered immune system signaling. Perturbation of the microbiota can lead to dysbiosis (i.e., a shift from the “normal” microbiota). If dysbiosis were occurring in male *An. stephensi* as a result of treatment with anti-*Asaia* mAb, it has the potential to trigger changes in immune system signaling, which could account for the observed transcript abundance patterns.

Male-specific genes were also enriched for genes involved in monooxygenase (GO: 0004497, p = 0.013) and oxidoreductase (GO:0016709, p = 0.021) activity (Appendix A). Biological process GO analysis revealed no significant enrichment after correcting for multiple comparisons, though the GO term “Proteolysis” (GO:0006508) and multiple GO terms related to metabolism were nominally significant (Appendix A, Appendix A). We found 11 genes regulated in male mosquitoes with a predicted implication in metabolism, corresponding to GO functions such as “carboxylic acid catabolic process,” “lipid metabolic process,” and “nucleoside metabolic process.” 

Of particular interest is the down-regulation of transcripts in males treated with anti-*Asaia* mAb related to metabolic pathways with multiple roles in insect physiology—an example is the transcript of the gene ASTE002146 encoding a putative scaffold family (PDZ-containing proteins) protein. These proteins are involved in the assembly of protein complexes in specific regions of the cell, allowing the organization of sub-cellular structures and signal transduction complexes [17]. Other examples are molecules like the cytochrome P450 monooxygenase (ASTE001621; ASTE009980) and the glucosyl/glucuronosyl transferase, as it is well known that there is cooperation between the insect host and its microbiota for detoxification [18]. Another down-regulated transcript (ASTE004882) in males injected with anti-*Asaia* mAb encodes a purine nucleoside phosphorylase. In insects, several examples of complementary purine metabolism between host and symbionts have been reported [19,20,21]. These examples suggest a possible role of *Asaia* in similar processes in adult mosquito males.

Multiple genes that were influenced by anti-*Asaia* mAb treatment are specifically expressed in tissues where *Asaia* is located (guts, salivary glands, reproductive organs). Some examples are the genes encoding several cytochrome P450s that, in the malaria vector *Anopheles gambiae*, are known to be expressed in the ovaries and in the male reproductive organs [22]. Genes encoding C-Type lectin/mannose binding proteins are expressed in salivary glands and/or midguts of some mosquito species [23,24]. Other genes, like serpins, are mainly expressed in mosquito midguts [25].

The generated transcriptomic data offer a repertoire of targets now known to be affected by *Asaia* presence that can be further investigated by RNA interference or alternative knock-down systems. It would be of interest to further explore tissue-specific patterns of gene expression, especially in tissues where *Asaia* shows particular tropism. This could be further expanded to female individuals and *Plasmodium* infection, in order to connect possible interactions with host vector competence.

Although our investigation is not able to conclusively demonstrate the symbiotic role of *Asaia* for its mosquito host, the observed sex-biased phenotype after anti-*Asaia* mAb injection and the contribution of the differentially regulated genes suggests a potentially broad and interconnected role of this symbiont in host physiology and survival.

## 3. Materials and Methods 

### 3.1. An. stephensi Rearing

*An. stephensi* Liston strain was maintained at standard 29 °C, 95% ± 5% relative humidity on a 12 h light/dark cycle. Adults were provided with water and 5% sucrose solution ad libitum. 

### 3.2. Production of Monoclonal anti-Asaia IgG

The monoclonal anti-*Asaia* IgG (anti-*Asaia* mAbs) was prepared following Kohler and Milstein methods [26]. Briefly, Balb/c mice were immunized using viable *Asaia* SF2.1 cells. The hybridoma was generated by the fusion of cells from murine myeloma cell line sp2/0-Ag-14 (ATCC, Rockville, MD, USA) with spleen cells from an immunized mouse. Handling of animals was performed under the control of the Ethical Committee of the University of Camerino, protocol number 2/2014, according to the Italian and European rules. Anti-*Asaia* antibody specificity was verified via enzyme-linked immunosorbent assay (ELISA) and immunostaining assays against various targets, including *Asaia* (SF2.1 strain), *Gluconobacter oxydans*, *Acetobacter aceti* (strains belonging to the Acetobacteraceae family) and *Escherichia coli* as an outgroup.

### 3.3. Immuno Fluorescence Assay (IFA) on *An. stephensi* Mosquitoes Microinjected with Anti-*Asaia* Antibody

Two cohorts of newly emerged *An. stephensi* mosquitoes were administrated with 5% sugar solution or 5% sugar solution plus *Asaia*-GFP respectively. At day 4 post-administration, mosquitoes were microinjected with anti-*Asaia* mAbs. After 24 h, 15 mosquito guts were dissected in PBS 1X and fixed with para 4% for 10 min at 4 °C. The organs were permeabilized with 0.5% Triton-100 in PBS 1X for 10 min at room temperature and washed three times for 10 min with 1X PBS. Samples were then incubated with anti-mouse IgG Alexa Fluor 594 (Invitrogen, Carlsbard, CA, USA) conjugated (1:100) in PBS 1X for 30 min at 37 °C and then washed for 10 min three times with PBS 1X. The organs were mounted with glycerol and visualized by confocal microscopy (Nikon C2plus) and NIS software (Nikon, Otawara, Japan).

### 3.4. Mosquito Survival Test

Five-day-old female and male mosquitoes (N = 30) were anesthetized on ice for intrathoracic injection. Glass capillaries (BF150-86-10, Sutter Instruments, Novato, CA, USA) were pulled using a Flaming/Brown Micropipette Puller System Model P-1000 (Sutter Instruments Company, Novato, CA, USA) to create microcapillary injection needles. Mosquitoes were injected with 69 nL of either 1.71 µg/µL anti-*Asaia* antibody diluted in PBS, 1.71 µg/µL Herceptin (Herc, IgG control) diluted in PBS, or PBS only (negative control) using a Drummond Nanoject II Automatic Nanoliter Injector (3-000-205-A, Drummond Scientific, Broomall, PA, USA) and glass microcapillary injection needles. Following injection, mosquitoes were maintained in the insectary under standard conditions, as described above, and their longevity was monitored. Data were gathered from three independent replicates and analyzed using the Kaplan–Meier method [27] in R (libraries “survminer”, “rms”, “survival”) (https://www.r-project.org/). 

### 3.5. Mosquito Transcriptome Analysis by Microarray Assay

Additionally, the transcriptome analysis was performed on female and male *An. stephensi* mosquitoes, 24 h post injection with anti-*Asaia* or Herc antibody as described above. Transcriptome analysis was performed using a custom Agilent microarray described previously [28]. The transcriptome analysis included four treatments, which were compared as follows: (i) female, Herc vs. (ii) female, anti-*Asaia*; (iii) male, Herc vs. (iv) male, anti-*Asaia*. For all treatments, four pools of 15 whole mosquitoes collected from the same experiment were used for RNA extraction. RNA was extracted from whole mosquitoes using the RNeasy Kit (Qiagen, Hilden, Germany) according to the manufacturer’s instructions. All samples were then treated with the Turbo DNA-free Kit (Invitrogen) according to the manufacturer’s instructions to remove genomic DNA. RNA was verified to be of high quality using an Agilent Bioanalyzer. All samples were labelled using the Two-Color Low Input Quick Amp Labelling Kit (Agilent Technologies, Santa Clara, CA, USA) according to the manufacturer’s instructions. Two hundred nanograms of RNA from each sample was used as input for the labelling reaction, and samples were labelled in a dye-swap design to prevent dye bias. Hybridization was performed according to Agilent’s Two-Color Microarray-Based Gene Expression Analysis Protocol. Feature extraction was performed using an Agilent Scanner and Agilent Feature Extraction Software. 

Microarray data analysis: Best hit gene IDs (E-value < 0/00001) were assigned to microarray probes using BLAST+ to locally query the AsteS1.2 *An. stephensi* gene set (downloaded 13 July 2015). Differential expression analysis was performed using limma in R [29] and as described in Short et al. (2017) [30]. Background correction was performed using the normexp method [31]. Signals within arrays were normalized using global lowess within-array normalization [32]. Gene Ontology enrichment analysis was performed using the GOstats package in R and Gene Ontology data for the *An. stephensi* genome from VectorBase.org (downloaded 17 March 2017). Gene Ontology pie charts were generated in Blast2GO using blastx to query the NCBI non-redundant database. 

## Figures and Tables

**Figure 1 pathogens-09-00380-f001:**
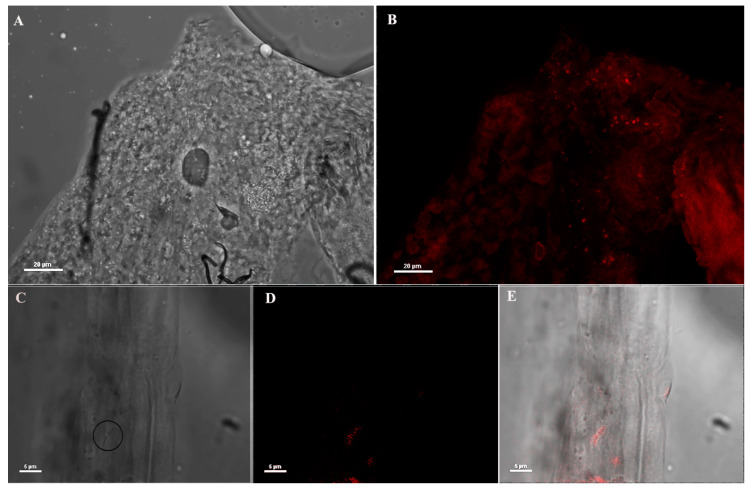
Immunofluorescence assay (IFA) on mosquito midguts using anti-*Asaia* monoclonal antibody (mAb). (**A**,**B**) Bright field microscopic and immunofluorescence images (40X) of male midgut using anti-*Asaia* mAbs; (**C**) different magnification (100X) of the tissues highlighting bacteria cells with the circle; (**D**) the corresponding immunofluorescent staining showing red-labelled bacteria recognized by the mAb; and (**E**) the superimposed image for localization. Scale bar = 20 μm (**A**,**B**) and = 5 μm (**C**–**E**).

**Figure 2 pathogens-09-00380-f002:**
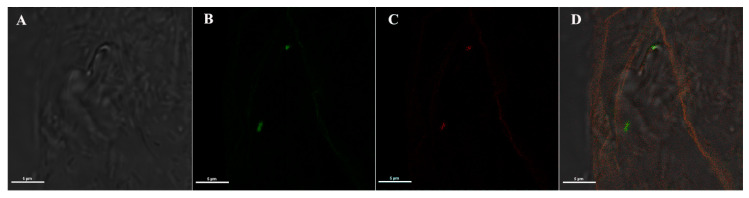
IFA on mosquito midguts colonized with *Asaia*-GFP using anti-*Asaia* monoclonal antibody. (**A**) Bright-field and (**B**) corresponding fluorescent images showing the green signal from an *Asaia* strain expressing the green fluorescent protein (*Asaia*-GFP); (**C**) red-labelled bacteria recognized by anti-*Asaia* mAbs, and (**D**) the co-localization of the signals on the merged image. Microscopy magnification is 100X and scale bar = 5 μm for all panels.

**Figure 3 pathogens-09-00380-f003:**
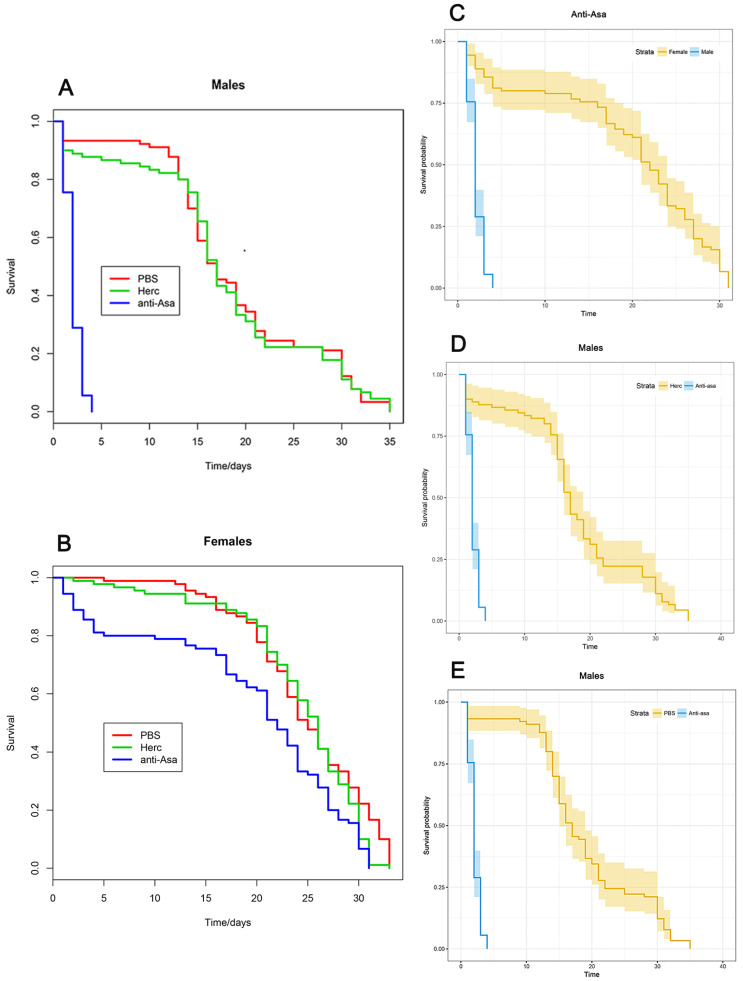
Longevity of *An. stephensi* male and female mosquitoes treated with anti-*Asaia* antibody. (**A**,**B**) Graphs show the survival rate of male and female mosquitoes treated with anti-*Asaia* antibody (blue), Herc antibody (green), and PBS (red). (**C**) Survival probability of male and female mosquitoes treated with anti-*Asaia* antibody. (**D**) Survival probability of male mosquitoes treated with anti-*Asaia* antibody vs. Herc antibody treatment. (**E**) Survival probability of male mosquitoes treated with anti-*Asaia* antibody vs. PBS treatment. Data represent the means of three independent replicates performed on different *An. stephensi* populations. Statistical analysis was performed using Kaplan–Meier methods in R showing high statistical significance (*p* < 0.0001) between anti-*Asaia* males and controls (**D**,**E**) and treated anti-*Asaia* males and females (**C**).

**Figure 4 pathogens-09-00380-f004:**
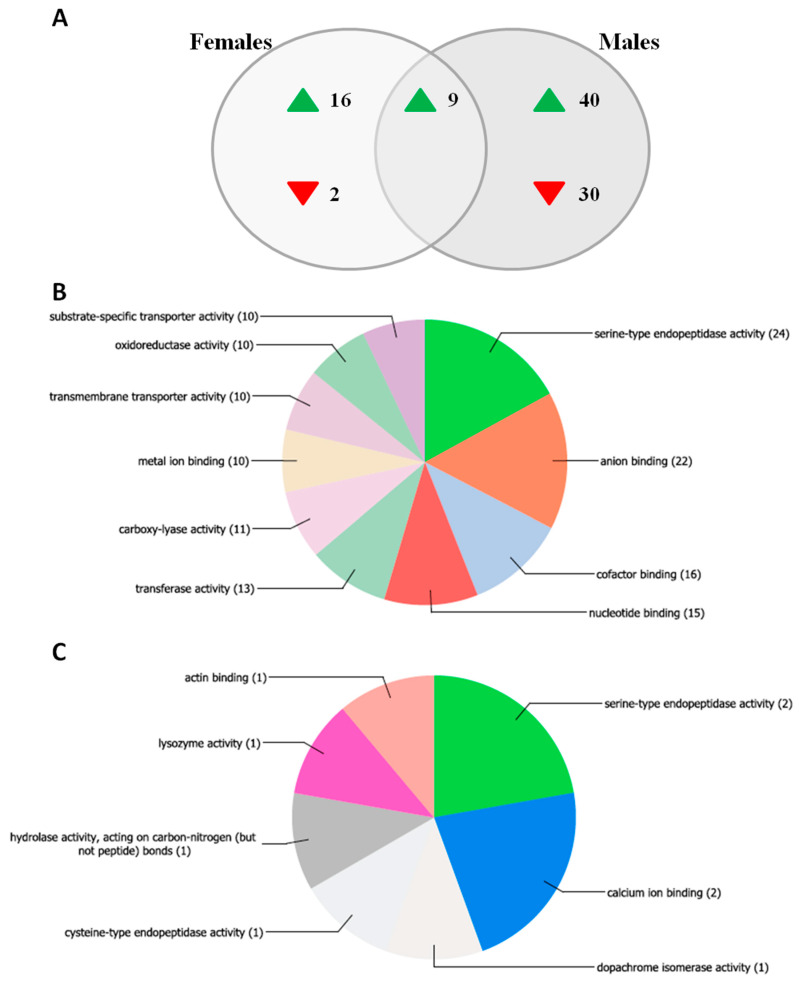
Transcriptomic analysis investigating the response to the anti-*Asaia* antibody in male and female *An. stephensi*. Male and female *An. stephensi* mosquitoes were injected with the anti-*Asaia* or Herc antibody, and the transcriptomic response of each sex to Anti-*Asaia* vs. Herc antibody treatment was assessed. (**A**) Venn diagram indicating the number of genes up- or down-regulated in males only, females only, or both males and females in response to anti-*Asaia* treatment. (**B**,**C**) Molecular function Gene Ontology (GO) distribution of transcripts affected by anti-*Asaia* treatment in a male- or female-specific manner. GO assignment for differentially regulated transcripts and chart generation was performed in Blast2GO. GO assignments are “multi-level,” meaning all terminal GO terms (i.e., terms at terminal nodes in the directed acyclic graph) for each dataset are included, regardless of GO level. Numbers in parentheses indicate the number of sequences assigned to each GO term; note that individual sequences can belong to more than one GO category.

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
