# Peer review of "Inhibition of Asaia in Adult Mosquitoes Causes Male-Specific Mortality and Diverse Transcriptome Changes"

_pathogens, 2020, doi:10.3390/pathogens9050380_

Round 1

Reviewer 1 Report

This paper entitled: “Inhibition of Asaia in adult mosquitoes causes male-specific mortality and diverse transcriptome changes” by Mancini et al. reports evidence on how treatment with a highly specific anti-Asaia monoclonal antibody impacts the survival and physiology of adult A. stephensi mosquitoes. The manuscript is well written and easy to follow.

The authors developed the specific anti-Asaia mAb and proved that it recognizes specifically Asaia bacterium (Asaia-GFP assay). Next, the mAb were injected to male and female mosquitoes to assess the influence of treatment on vector’s physiology, i.e., lifespan. Also, profound changes in transcriptome of males and females was found upon mAb treatment, pinpointing sex-differentiated regulation of gene expression program.

This reviewer has the following suggestions to the authors:

  • I would move both supplementary figures related to the immunofluorescence analysis of anti-Asaia mAb and bacterial colocalization to the main body of the manuscript, as they are essential for estimating the mAb specificity.
  • More detailed description of the transcriptome analysis is needed. Did the authors check RNA integrity? How many probes did the microarray consisted of? What type of normalization was performed?
  • Moving forward, tissue-based expression profiling would be the next step in understanding the influence of Asaia on gene functions of  Stephensi. Also, the analysis of gene expression data in the context of changes due to blood meal (in females) and infection might lead to new perspectives and insights in vector–pathogen interaction and disease transmission. The authors should address their future directions.

Author Response

We would like to thank the reviewer for its constructive comments and suggestions for improving our manuscript. In the following, we highlight and address concerns of reviewers and their specific comments.

Rev1

I would move both supplementary figures related to the immunofluorescence analysis of anti-Asaia mAb and bacterial colocalization to the main body of the manuscript, as they are essential for estimating the mAb specificity.

We thank the reviewer for the suggestion. We merged and added the immunofluorescence images on midguts as Fig.1 in the main text. We moved also Fig. S2 into the main text as Fig. 2.

More detailed description of the transcriptome analysis is needed. Did the authors check RNA integrity? How many probes did the microarray consisted of? What type of normalization was performed?

RNA was verified to be of high quality using an Agilent Bioanalyzer. The construction and contents of the microarray is described in detail in reference 28 (Pike et al., 2014). Best hit gene IDs (E-value < 0/00001) were assigned to microarray probes using BLAST+ to locally query the AsteS1.2 An. stephensi gene set. Differential expression analysis was performed using limma in R as described in Short et al., 2017. Signals within arrays were normalized using global lowess within-array normalization.

We have expanded the related part in the Material and Methods section and included a more detailed description of the transcriptomic analysis.

Moving forward, tissue-based expression profiling would be the next step in understanding the influence of Asaia on gene functions of  Stephensi. Also, the analysis of gene expression data in the context of changes due to blood meal (in females) and infection might lead to new perspectives and insights in vector–pathogen interaction and disease transmission. The authors should address their future directions.

We have now addressed the suggested future perspectives in the discussion section.

Reviewer 2 Report

In this study, the authors sought to characterize the fitness benefits of Asaia bacteria in Anopheles stephensi mosquitoes to further assess the potential of Asaia as a vector-control strategy. To do so, they attempted to employ loss-of-function (through inhibition of Asaia) and transcriptomics. While the premise of their study is interesting and of potential value, I feel that the study as a whole was poorly controlled, and as presented, does not definitively support the conclusions the authors make. If the authors are willing to perform additional experiments to address the concerns given below, and if those experiments alleviate these concerns, the manuscript may be suitable for publication.

1) In figure S2, it is clear that there is some autofluorescence in the red channel in panel B (this is to be expected). However, the image that detects the antibody in the red channel (Panel C) appears to have markedly more signal distributed diffusely throughout the midgut than panel B…is this non-specific labeling? To me it looks like it is and calls into question the assumption that this antibody does not exhibit any reactivity with endogenous targets.

2) Moreover, even if one accepts the results from Figures S1 and S2 as demonstrating specificity to Asaia, it is only done so in the midguts of the mosquitoes. Since the antibody is injected into the hemocoel, it will be distributed systemically and could very well have off target effects in tissues other than the midgut, which may lead to the differences observed in survival. Indeed, the sex-specific mortality observed may actually be explained by the anti-Asaia mAb targeting an endogenous antigen specific to males.

3) As a control for the experiment showing differences in survival between mosquitoes injected with anti-Asaia mAb or PBS/irrelevant antibody controls, an identical experiment using axenic (bacteria free) mosquitoes should have been performed in tandem. While a direct comparison between the survival times of axenic and normal mosquitoes may not be able to be made, one could see if the effect observed in normal mosquitoes (i.e. sex-dependent decreases in survival times between mosquitoes injected with anti-Asaia mAb vs. controls) was replicated in axenic mosquitoes. If the same result was observed in axenic mosquitoes, it would imply that the differences in survival are not dependent (at least entirely) on neutralization of Asaia by the mAb, and instead point to an off-target effect. Strategies for generating axenic mosquitoes that exhibit complete development into adults have been described (Correa et al 2018, Nature Communications PMID 30367055).

4) Given the concerns above, I have problems accepting the conclusions from the transcriptome sequencing at face value.

Minor comments:

1) The image quality of Figure S1 is poor (at least on my computer screen); it is very difficult for me to visualize the red channel.

Author Response

We would like to thank the reviewer for its constructive comments and suggestions for improving our manuscript. In the following, we highlight and address concerns of reviewers and their specific comments.

1) In figure S2, it is clear that there is some autofluorescence in the red channel in panel B (this is to be expected). However, the image that detects the antibody in the red channel (Panel C) appears to have markedly more signal distributed diffusely throughout the midgut than panel B…is this non-specific labeling? To me it looks like it is and calls into question the assumption that this antibody does not exhibit any reactivity with endogenous targets.

After Rev.1’s suggestion, Fig. S1 and S2 were compiled and moved into the main text. We adjusted the background autofluorescence parameters that, as mentioned, are fairly expected with assays on tissues using red channel fluorescence; in the supplementary materials, we have now included an additional immunofluorescence image from the same assay performed on female guts and ovaries. A comparable background autofluorescence can be detected also in these tissues, whilst stronger fluorescent signal from the MAb can be found in relation to Asaia cells. We believe that these additional images will support and add insights to demonstrate antibody specificity and undermine the possibility of an off-target binding on male host antigens.

2) Moreover, even if one accepts the results from Figures S1 and S2 as demonstrating specificity to Asaia, it is only done so in the midguts of the mosquitoes. Since the antibody is injected into the hemocoel, it will be distributed systemically and could very well have off target effects in tissues other than the midgut, which may lead to the differences observed in survival. Indeed, the sex-specific mortality observed may actually be explained by the anti-Asaia mAb targeting an endogenous antigen specific to males.

Tissues were chosen on the basis of Asaia tropism: previous studies demonstrated that Asaia almost uniquely naturally colonizes midguts and reproductive organs of An. stephensi. As anticipated, we have now included images from IFA performed on female tissues, to show that, at least in the analysed tissues, the signal and co-localization with Asaia cells is comparable between males and females, even though they show different survival phenotypes. Moreover, due to the nature of the immunofluorescence assay used to demonstrate antibody specificity, the analysis of whole mosquito carcass or of the hemocoel would have been very challenging to carry out.

3) As a control for the experiment showing differences in survival between mosquitoes injected with anti-Asaia mAb or PBS/irrelevant antibody controls, an identical experiment using axenic (bacteria free) mosquitoes should have been performed in tandem. While a direct comparison between the survival times of axenic and normal mosquitoes may not be able to be made, one could see if the effect observed in normal mosquitoes (i.e. sex-dependent decreases in survival times between mosquitoes injected with anti-Asaia mAb vs. controls) was replicated in axenic mosquitoes. If the same result was observed in axenic mosquitoes, it would imply that the differences in survival are not dependent (at least entirely) on neutralization of Asaia by the mAb, and instead point to an off-target effect. Strategies for generating axenic mosquitoes that exhibit complete development into adults have been described (Correa et al 2018, Nature Communications PMID 30367055).

We appreciate the rationale behind the request of the reviewer and we agree with it; in fact, the creation of an axenic An. stephensi line to be used as a control for ruling out off-target effects was included in the original design of these experiments. Although, multiple protocols were tested without success, including the suggested Correa et al 2018. We think it is useful to underline that such protocol was optimized on Ae. aegypti mosquitoes and it could have a stronger impact on fitness cost of Anopheles mosquitoes, which is what we have actually seen. The different antibiotic treatments followed by the intra-thoracic injection step resulted in a significant increased mortality rate in both mosquito sexes, which critically impacted on the numbers of mosquitoes for the experiment and imposed an additional bias on the phenotype we aimed to assess.

Unfortunately, replicating the whole set of experiments including the axenic line, would be very complex for various reasons and logistically challenging due to the public health emergency we are experiencing at the moment with the consequent lockdown which is a great impediment to further experimental activities.

4) Given the concerns above, I have problems accepting the conclusions from the transcriptome sequencing at face value.

We think that the clarifications about the immunofluorescence images on the specific recognition of the mAb against Asaia would lead to a more clear interpretation of the transcriptomic data.

Minor comments:

1) The image quality of Figure S1 is poor (at least on my computer screen); it is very difficult for me to visualize the red channel.

The image quality has been improved; it can be also found as a .zip file.

Round 2

Reviewer 2 Report

The authors acknowledge the value of two of my critiques (and state that they even planned to include one as a control initially, but had to abandon it due to technical difficulties), but fail to actually address them. While I understand that additional experiments are always difficult to incorporate into manuscript revisions, and even moreso given the current situation with SARS-CoV-2, that does not undermine the importance of those experiments, which again, by their own admission, the authors state they initially planned to perform.

As a compromise, I would strongly suggest the authors tone back the strength of their conclusions, as I am not entirely convinced what they are concluding is real, at least as presented.

Author Response

As strongly suggested by the reviewer we have toned back the strength of our conclusions.

Round 3

Reviewer 2 Report

I am satisfied with the changes the authors have made to the manuscript.